# Probation as a targeted entry point for scaling up opioid agonist therapies in Moldova using a modified screening, brief intervention and referral to treatment strategy

Matthew N. Ponticiello[1,2], Daniel J. Bromberg[1,3], Lyu Azbel[1,3], Sergiu Cugut[4], Svetlana Doltu[4], Frederick L. Altice[1,2,3]*

1 Yale School of Medicine, New Haven, Connecticut, United States of America, 2 Yale School of Public Health, New Haven, Connecticut, United States of America, 3 Center for Interdisciplinary Research on AIDS, Yale School of Public Health, New Haven, Connecticut, United States of America, 4 AFI (Act for Involvement), Chisinau, Republic of Moldova,

* frederick.altice@yale.edu

## Abstract

People with HIV and opioid use disorder (OUD) are concentrated in criminal legal settings. Consequently, decarceration efforts in Moldova have shifted many people in prisons to community supervision in probation. Probation differs from prisons as they have no mandate to screen and treat for medical conditions like OUD and link to evidence-based treatments like methadone or buprenorphine, otherwise known as medications for opioid use disorder (MOUD). We hypothesized that probation settings are a missed opportunity to screen and link to treatment for HIV and OUD. From November 2019 to April 2023, 900 consecutive entrants to 10 probation centers across Moldova underwent a modified screening, brief intervention, and referral to treatment (SBIRT) strategy to identify people on probation with OUD and link them to MOUD; testing for HIV, HCV, HBV, and syphilis was bundled for efficiency. Among those screening positive for OUD, interest in MOUD was assessed before and after the brief intervention. For those interested in MOUD, they were referred to treatment. MOUD interest was assessed using a single-item screening instrument, where higher scores were associated with greater interest in starting MOUD. Overall (N=900), 15.1% (N=136) participants met screening criteria for OUD, of which 119 (87.5%) were eligible for and enrolled in the brief intervention. Interest in MOUD after the brief intervention increased modestly, but significantly (p<0.001). Overall, 33 (27.7%) of 119 participants initiated MOUD, and 32 (97%) of these were retained on treatment over 6-months. Predictors of initiating MOUD were perceiving MOUD to be an important treatment for OUD and having HCV and HIV. Twenty-seven (22.7%) tested positive for HIV, with 7 (25.9%) being newly diagnosed during the screening procedures. Rapid screening strategies can identify and provide opportunities for treatment for multiple interrelated chronic conditions. Implementation opportunities, however, will require aligning public safety mandates in probation with public health.

**Data availability statement:** Data has been made available on the repository Dryad: DOI: 10.5061/dryad.573n5tbq3.

**Funding:** This work was supported by the National Institutes of Health: R01 DA029910, FLA and F31DA054861, DJB (National Institute for Drug Abuse); F30AI198064, the Global Health Emerging Scholars Program under award number D43TW010540 MNP (National Institute for Allergy and Infectious Diseases and Fogarty International Center along with additional support of by the Office of AIDS Research); and institutional grants T32 MH020031 LA and T32 GM13665 MNP. The funders had no role in study design, data collection and analysis, decision to publish, or preparation of the manuscript.

**Competing interests:** The authors have declared that no competing interests exist.

## Introduction

Medications for opioid use disorder (MOUD) include naltrexone, buprenorphine and methadone. Only opioid agonists like methadone and buprenorphine, however, substantially reduce crime, incarceration, overdose, and mortality and increase employment and social functioning [1]. For individual and public health, however, initiating and remaining on MOUD with these opioid agonists is inherently prosocial. MOUD improves an individual's health and functioning while also generating broad societal benefits by reducing overdose mortality, destabilizing cycles of withdrawal-driven injecting, and lowering risks of HIV and HCV transmission. Through these pathways, it can also reduce crime, improve employment and relationships, and strengthen community safety [2–4]. The combination of these benefits results in improved health-related quality of life and general social functioning [5]. Due to the inextricable link between non-medical opioid use and crime, criminal legal settings concentrate people with OUD, HIV and HCV [6–8] and serve as potential entry points for screening and treatment [8]. While most screening and treatment models in the criminal legal settings have focused on this strategy in prisons and pre-trial detention centers [9], community supervision settings like probation, where people live freely in the community, are likely to interact with large numbers of people with OUD [10]. These probation "touchpoints", however, have not been adequately deployed as settings to align public safety and health. This is especially true as people on probation, in the absence of MOUD, have substantially heightened rates of overdose-related mortality and criminal activity [11,12].

As many settings align drug policy and public health [13], there has been a recent shift globally to reduce incarceration levels in prisons and jails, in part by shifting people with non-violent and less severe crimes to community-based supervision, like probation. This shift represents an intermediary step on the path to decarceration efforts, potentially resulting in increasing numbers of people with or at risk for OUD, HIV, HCV, and tuberculosis to be concentrated in probation settings where there is a mandate to ensure public safety, yet not for public health. Probation settings, however, share a common mandate by addressing OUD, which, when untreated, undermines public safety efforts. For example, when OUD is left untreated, it is associated with higher levels of criminal activity [14]. Moreover, as probation settings also lack infrastructure and staffing to independently align both public safety and public health, it is crucial to identify unique opportunities for integrating such activities.

The Eastern European and Central Asian (EECA) region includes countries with some of the highest prevalences globally of OUD, HIV, HCV, tuberculosis, as well as high incarceration rates [5,6,15–18]. This is especially true as HIV incidence and mortality in EECA continue to increase among people who inject drugs (PWID), with high levels of people with HIV (PWH) in this region [19] who are concentrated in criminal legal settings due to the criminalization of drug use [20]. The rising incidence of HIV among PWID, coupled with the increased concentration of PWID/PWH in criminal legal settings suggest the need for expanding opportunities to scale-up HIV prevention using MOUD to achieve epidemic control. MOUD is an evidence-based primary and secondary [21,22] HIV prevention strategy and is key to reducing transmission and mortality [23]. It also improves treatment outcomes for HCV [24–26], and tuberculosis

[27]. Moreover, when scaled up, MOUD can significantly reduce HIV incidence and mortality [1,6,23], and is particularly effective at preventing HIV and transmission among criminal legal-involved individuals with OUD [27,28]. Most MOUD programs in EECA, aside from Ukraine, which has been scaling up substantially since 2014 [29,30], remain under-scaled and far below targets set by the World Health Organization [31].

Moldova is a lower middle-income country (LMIC) in EECA with a population of approximately three million people, with about a third of people living in rural regions [32]. Across the country, including in both urban and rural settings, the treatment cascade for OUD and HIV remains suboptimal. HIV incidence and mortality continue to increase in Moldova, unlike patterns globally, partly due to suboptimal scale-up of HIV prevention programs [33]. Among PWID, opioid injection remains common, and HIV prevalence is elevated, while MOUD coverage remains far below population need. Structural barriers, including stigma, criminalization, and administrative restrictions that shape daily functioning and employment, further constrain uptake and retention [34,35]. Together, these conditions create a compelling rationale to test whether probation can serve as a scalable entry point to identify OUD and facilitate linkage to MOUD as an HIV and HCV prevention strategy.

PWID in community settings rarely utilize conventional primary care and specialty services [36], in part due to high levels of stigma in such settings [37–39]. Instead, they are more likely to intermittently rely on other touchpoints for care like syringe exchange programs, emergency departments, and prisons or jails [40], rather than engage in long-term care. Criminalization of drug use, stigma, and discrimination, however, remain major barriers to care for HIV prevention services among PWID [41]. In Moldova, coverage of HIV prevention services among PWID remains suboptimal, and MOUD remains underutilized relative to need.

As part of a larger research study, we pilot-tested a similar protocol for both Ukraine and Moldova. The procedures for Ukraine have previously been described to assess the deployment of a modified screening, brief motivation and referral to treatment (SBIRT) introduced in probation settings [42]. The modification included rapid screening using a single-item screening question, evaluation for treatment, and treatment initiation (SET) by greenlighting a treatment pathway. This study was conducted in parallel with the Ukrainian one [43]. The primary goal of this study was to assess the feasibility of probation as a clinical opportunity to identify people with OUD (and HIV and HCV) from a syndemic perspective [44] and facilitate linkage to MOUD using an externally delivered, modified SBIRT strategy. In this study, feasibility was defined as the ability to: 1) systematically screen, evaluate and treat (SET) consecutive probation clients for OUD; 2) deliver a brief intervention in the probation setting without disrupting routine operations; and 3) successfully link interested individuals to MOUD, with verification of treatment initiation and retention. As a pilot study, it did not assess implementation outcomes such as adoption by probation staff, long-term sustainability, or costs. There is a dearth of literature on the use of modified SBIRT for substance use disorders, particularly in settings where MOUD uptake is suboptimal and HIV prevalence is high. We hypothesized that a modified SBIRT strategy would increase interest in MOUD among people on probation in the Moldovan setting, and thus MOUD uptake. Additional details on Moldova's epidemiologic profile, incarceration and probation landscape, and the availability and financing of MOUD and syringe services programs are provided below.

## Methods

### Ethics statement

This study is registered at www.clinicaltrials.gov (NCT 04947475) and was approved by the institutional review boards at Yale University and the Moldovan Institute of Pthisiopneumology Chirl Dragniuc.

### Moldovan context

Moldova is an upper middle-income country in EECA with approximately 3 million people. The criminal legal system supervises a large population in the community, in addition to those incarcerated. In 2025, there were 5,892 people in prison, translating to an incarceration rate of 255 people per 100,000 [45]. In 2023, there were 320 people per 100,000 population on probation [46]. In this context in 2018, 1,990 people in prisons received risk reduction services at 33 syringe

exchange points in 13 penitentiary institutions [47]. In Moldova, only 61.7% of PWID were covered by HIV prevention services. Despite this, at the end of 2022, MOUD was available at 10 locations in the civil sector and 13 in penitentiary institutions [48]. Both methadone and buprenorphine are available in Moldova for free, including within penitentiary settings. There are currently 13 MOUD sites in prisons and 9 community-based sites, mostly integrated within narcological (addiction treatment) services, treatment is offered free of charge through the public health sector. In this study, the only MOUD available are opioid agonist therapies (OAT) including methadone or buprenorphine. Moldova's Ministry of Health provides MOUD for free in prisons and within the community, including publicly available methadone and buprenorphine. The procurement of medications is financed from the state budget, while the psychosocial support component remains largely donor-funded and implemented by NGOs. Similarly, syringe services programs are available in all prisons and pre-trial detention centers and include delivery by professionals, peers and through vending machines [49].

## Study setting

This study was conducted at 10 probation centers in all three regions in Moldova between November 1, 2019 and April 30, 2023. Sites were selected based on proximity to MOUD sites. To avoid potential conflicts between the officer's supervisory role and the research procedures, the research team conducted study activities in the same office where the officer typically meets clients every 2–4 weeks, but all research procedures were conducted in a private room by the research team, independent of probation staff. This room was available for initial screening and brief intervention, though follow-up assessments could be conducted elsewhere privately in either in a research setting, the participant's home or another confidential location agreeable to the client.

In 2010, Moldova established the National Probation Inspectorate to decrease the prison census, prevent recidivism, and to reintegrate people on probation into the community. The probation system combines 38 probation offices [50] to oversee persons deemed stable enough to be supervised in community settings. It is linked to social rehabilitation centers that supplement these services in three territorial regions: north, center and south. At the end of 2018, there were 11,970 people on probation [51]. Unlike in prisons, the prevalence of OUD among people on probation or the number of people who receive long-term treatment for HIV, tuberculosis, or OUD is unknown.

Moldova's probation service operates under Law No. 8 since February 14, 2008, which outlines the organization, mandates, and scope of probation services, including their responsibilities in rehabilitation and oversight [52]. These services are independent of prisons, and all are delivered exclusively in community settings, enabling probation officers to engage flexibly with family networks, community health and social services, and employment opportunities. The central National Probation Inspectorate (NPI) oversees 38 territorial probation offices, which are organized into three regional inspectorates in the North, Centre, and South regions [53]. Among these offices, only three include "centers for social rehabilitation." These centers are strictly administrative and serve as daytime locations for delivering probation-related rehabilitation programs, not as detention facilities.

Individuals under probation may include: persons undergoing pre-trial psychosocial evaluations; those conditionally or post-released from criminal sentences (both individuals released from prison before completing their sentence and those sentenced directly to non-custodial supervision); individuals serving unpaid community service; and individuals subject to protective or family-related court orders. Placement on probation is determined by the court, based on the case facts and aligned with the Criminal Code's provisions. For example, unpaid community service may be imposed as a standalone community sanction or as part of a suspended sentence.

Moldova's probation service includes 305 staff, including probation officers who must have a bachelor's degree or equivalent in law, psychology, or social work. The daily caseload for each of the offices ranges from 44 and 1612 clients with an average of 58 clients per officer; this is almost twice higher than in other EU states (mean = 33) [54].

Probation services have traditionally prioritized public safety over health-related interventions. Probation officers typically lack legal authority to inquire about health conditions unless clients voluntarily disclose them; in such cases, officers may

provide referrals to services. While probation can legally inquire about and test for substance use, a recognized health issue, clients may feel disinclined to disclose usage, of concern that officers may view this as a probation violation and refer them for sanctions. This tension highlights an inherent conflict: substance use and dependence are both public health and public safety issues and would benefit from an integrated approach. The challenge lies in aligning these domains, finding an integrated strategy that supports individual health while maintaining societal protection. Recent approaches offer promising models at this intersection. For instance, integrating community health workers (CHWs) with probation services has been shown to alleviate officer workload and build trust, while improving client access to treatment and reducing burnout for officers - an example of a public health–public safety bridge [55].

Similarly, probation has been recognized as a crucial public health touchpoint for identifying individuals with OUD and linking them to OAT. Interventions such as SBIRT (Screening, Brief Intervention, and Referral to Treatment) models in probation settings demonstrate the feasibility of pairing health screening with public safety supervision [43].

To enroll on MOUD, it is required that a person with OUD be diagnosed by an addiction treatment specialist and placed on a national registry. Once on this registry, and while on MOUD, a person's personal liberties are restricted as they are ineligible for a driver's license and restricted from some types of employment (e.g., operating heavy equipment). This national registry, much like throughout all of EECA, is a major impediment to MOUD enrollment [56,57,41,58–60].

## Participants

All clients attending probation visits were consecutively approached for screening. Because enrollment required participants not be currently receiving MOUD, baseline MOUD initiation was zero by design; therefore, analyses of MOUD initiation are descriptive and not comparative. While in the waiting room, the research assistant would invite them for a brief assessment of opioid use using a single-item screening question (SISQ) [61–63]. OUD diagnosis was confirmed using the brief Rapid Opioid Dependence Screen following the positive single-item screen [64]. Anyone screening positive was then consented for further participation, including for the brief intervention and referral to treatment. Inclusion criteria included being under probation supervision and: 1) 18 years or older; 2) screened positive for OUD; 3) live within 30 kilometers of an MOUD site; and 4) have not received MOUD for at least 15 days. Participants were given the opportunity to opt-out of the study at any time without it impacting the medical care they were offered. Individuals who were not interested in initiating MOUD at the time of screening were provided with a standardized packet with referral information for harm reduction and other relevant health services available through NGOs and medical units. This included details on needle and syringe programs, HIV testing and care, psychosocial support, and MOUD services for those who may wish to engage in treatment later. A comprehensive directory of harm reduction centers operating in Moldova was developed and shared with probationers. The document outlines health risks associated with opioid dependence and provides guidance on accessing community-based services for drug treatment, social support, HIV and HCV testing, and other health services. The information package also includes a list of all community-level health and harm reduction services available nationwide.

## Procedures

Trained research assistants, not probation officers, screened consecutive entrants scheduled to see the probation officer using modified SISQ for OUD, which is 86% sensitive [63]. For those who screened positive, participants underwent written informed consent procedures to allow for evaluation, which included confirmation of OUD using the brief Rapid Opioid Dependence Screen [65]. After consent, participants completed the baseline survey and brief intervention. All interviews were self-administered via REDCap on tablet computers. Following the survey, the research assistant delivered the 10-minute brief intervention using motivational interviewing techniques. Both methadone and buprenorphine were offered as treatment if a participant qualified for participation. Staff were able to link study participants to patient records through a unique registration code in MOUD provided to the research team by the study participant. All research assistants had a bachelor's degree at minimum and were trained on the study protocol, and intervention. Motivational interview training was

conducted by two psychologists, a psychiatrist, a lawyer and a social worker with experience working with people in the criminal legal. Interviews were conducted in Romanian or Russian, depending on participants' preference.

The brief intervention included risks of opioid use by illustrating the health consequences as well as the relative benefits of MOUD (e.g., HIV risk reduction, criminalized substance use, prevention of overdose) compared to continued opioid injection as part of the intervention. Research assistants were provided coaching and the sessions were checked for quality and fidelity by the study coordinator. Interest in MOUD was assessed both before and after the brief intervention (at baseline). Anyone interested in MOUD was offered immediate linkage to MOUD. All follow-up visits were coordinated between the research assistant and probation personnel based on scheduled visits and participant legal status.

Participants completed a second interview and another brief intervention approximately 30 days after the initial intervention. There were no limitations on MOUD initiation and for anyone who initiated MOUD, they were followed for 6 months from baseline (and the SBIRT) to assess retention in MOUD. Each interviewer maintained monthly contact with the MOUD site to track inclusion in MOUD and update the data about participant medical status. The equivalent of 10 USD in Moldovan lei was provided as compensation for participants' time interviewing.

### Brief Intervention (BI)

The brief intervention could last up to 30 minutes, allowing time for participants to ask questions. The participants met one-on-one with a trained interviewer to get information under the "brief intervention" component of the research. The brief intervention has four objectives:

1. Identify the level of interest in MOUD, to focus on elements critical in motivational interviewing [43,66].

2. To address information deficits related to the dangers of psychoactive substance use by informing them about the potential hazards and negative health effects with ongoing use and the benefits of MOUD.

3. To motivate the participants to reduce substance use and related risk behaviors, including the use of MOUD and syringe services programs.

4. To support behavioral skills to access risk reduction services.

After the brief intervention, the researcher reassessed the interest of the participant to start the MOUD. If the participant accepted MOUD, each participant was interviewed at baseline (Day A) and followed up at 1, 3, and 6 months by the same interviewer, who tracks timelines in Tracking Form 1. After Day A, interviewers maintain contact with participants and probation officers to schedule interviews within the window period (two weeks before and after the target date). Missed or delayed interviews and reasons are documented. At each follow-up, interviewers conducted a REDCap interview on tablet/computer and verify OAT enrolment or assess interest in OAT. All data are linked to unique participant codes to ensure confidentiality. Interviewers receive monthly online training and coordinate with probation staff and the research team to update participant status, including any detention or deregistration.

### Laboratory procedures

All consented participants with OUD underwent rapid, point-of-care testing for HIV, HBV, HCV, and syphilis at baseline and after 6 months. NGO-based harm reduction service providers conducted rapid testing and counseling or assisted the self-testing. Probation staff facilitated referrals, while testing was performed on-site at the probation office. HIV and HCV screening using rapid antibody tests. Individuals who tested HCV antibody–positive were further referred to local clinics for confirmatory HCV RNA testing through the national public health laboratory network. Syphilis testing was performed using rapid plasma reagent (RPR) screening test. At baseline and at months 1, 3, and 6 participants underwent urine drug testing for opioids.

## Survey measures

MOUD initiation was evaluated within 6 months of the baseline interview. Initiation dates were verified using Moldova's national database for MOUD monitoring. Interest in, difficulty in accessing, and importance of initiating MOUD were measured before and after the brief intervention. All three measures, interest in MOUD, perceived difficulty in accessing MOUD, and perceived importance of initiating MOUD, were each rated from 0 to 10, with higher scores indicating greater levels of the construct assessed (e.g., higher importance, greater perceived difficulty, or higher interest).

Other standardized measures included an 8-item, 5-point Likert scale that assessed MOUD knowledge and attitudes [67] with higher numbers indicating greater knowledge and more positive attitudes. Two items measured attitudes towards MOUD (Cronbach's α = 0.74) while six items related to participants' knowledge (Cronbach's α = 0.93) of MOUD on health and criminal activity outcomes. We also used a 5-point Likert scale that assessed perceived effectiveness of various treatments. Of note, for the purpose of surveys, opioid substitution therapy (OST) was used rather than MOUD due to more common use of this term in EECA. The Center for Epidemiological Studies-Depression (CES-D)'s 10-item scale was used to asses for depression where scores ≥10 were considered moderate to severe depression [68]. Drug-related problems were measured using the validated 10-item Drug Abuse Screening Test (DAST-10), with scores ranging from 0–10; higher scores reflect greater severity of drug-related consequences. Social support was measured using a brief 5-item scale where higher scores corresponded to higher perceived social support.

## Statistical analysis

The primary outcome was initiation of MOUD, defined as documented receipt of methadone or buprenorphine during the study follow-up period. Secondary outcomes included time to MOUD initiation and changes in self-reported interest in MOUD after the brief intervention. Analyses were designed to address three related scientific questions: 1) What individual- and system-level factors are associated with MOUD initiation among people on probation; 2) What level of self-reported interest best discriminates subsequent MOUD initiation; and 3) How does MOUD interest change after the brief intervention, and how is this related to treatment uptake?

To address the first question, we used multivariable logistic regression to examine associations between baseline characteristics and MOUD initiation. Covariates were selected *a priori* based on theory and prior literature. To address the second question, ROC analysis was conducted to evaluate the ability of baseline MOUD interest (0–10 scale) to predict subsequent MOUD initiation. Documented MOUD initiation served as the binary outcome. The optimal cutoff point was identified using the Youden index and applied in subsequent regression analyses. To address the third question, changes in MOUD interest were examined descriptively and using mixed-effects models to account for repeated measures within individuals. MOUD interest scores were summarized using medians and interquartile ranges (IQR). Changes in paired interest scores before and after the brief intervention were evaluated using the Wilcoxon signed-rank test. All statistical tests were two-sided with a significance level of α = 0.05. Analyses were conducted using STATA 18.0 (College Station, Texas). $X^2$ and Fisher's exact test were used to compare categorical variables as appropriate. T-test or Mann Whitney U test were used to compare continuous variables as appropriated. Shapiro-Wilk test was used to test for normality. Two logistic regression models were built for predictive variables of MOUD initiation. One with bivariate associations of MOUD initiation, and a multivariable logistic model was analyzed using backward selection.

Receiver operating characteristic (ROC) analysis was used to evaluate the ability of participants' initial self-reported interest in MOUD (measured on a 0–10 scale at baseline) to predict subsequent initiation of MOUD during follow-up. The purpose of the ROC analysis was not hypothesis testing but to establish a clinically interpretable threshold on a continuous motivation scale to support downstream modeling and interpretation. The binary outcome ("gold standard") for the ROC analysis was documented MOUD initiation (yes/no), defined as verification of receipt of methadone or buprenorphine through treatment program confirmation. ROC analysis was conducted using the baseline MOUD interest score only. Although interest was assessed after the brief intervention, only the initial score was used for this analysis to preserve

temporal ordering between predictor (interest) and outcome (MOUD initiation) and to avoid correlated observations within individuals. The area under the ROC curve (AUC) was calculated to assess discrimination, and the optimal cutoff point was determined using the Youden index to maximize sensitivity and specificity. The identified cutoff was then used to dichotomize interest level for subsequent regression analyses. Mean values were presented descriptively for interpretability, while statistical significance for paired differences was determined using the Wilcoxon signed-rank test.

## Results

### Participant disposition

Of the 900 consecutive entrants screened, 136 (15.1%) met criteria for OUD. Of these, 17 did not meet study eligibility because they did not live within 30 kilometers of an MOUD site, resulting in 119 (100%) who enrolled in the study, all of which completed the brief intervention (Fig 1). These findings demonstrate the feasibility of identifying individuals with OUD and linking them to treatment through probation-based screening, even in the absence of a formal healthcare mandate. Participant characteristics are shown in Table 1.

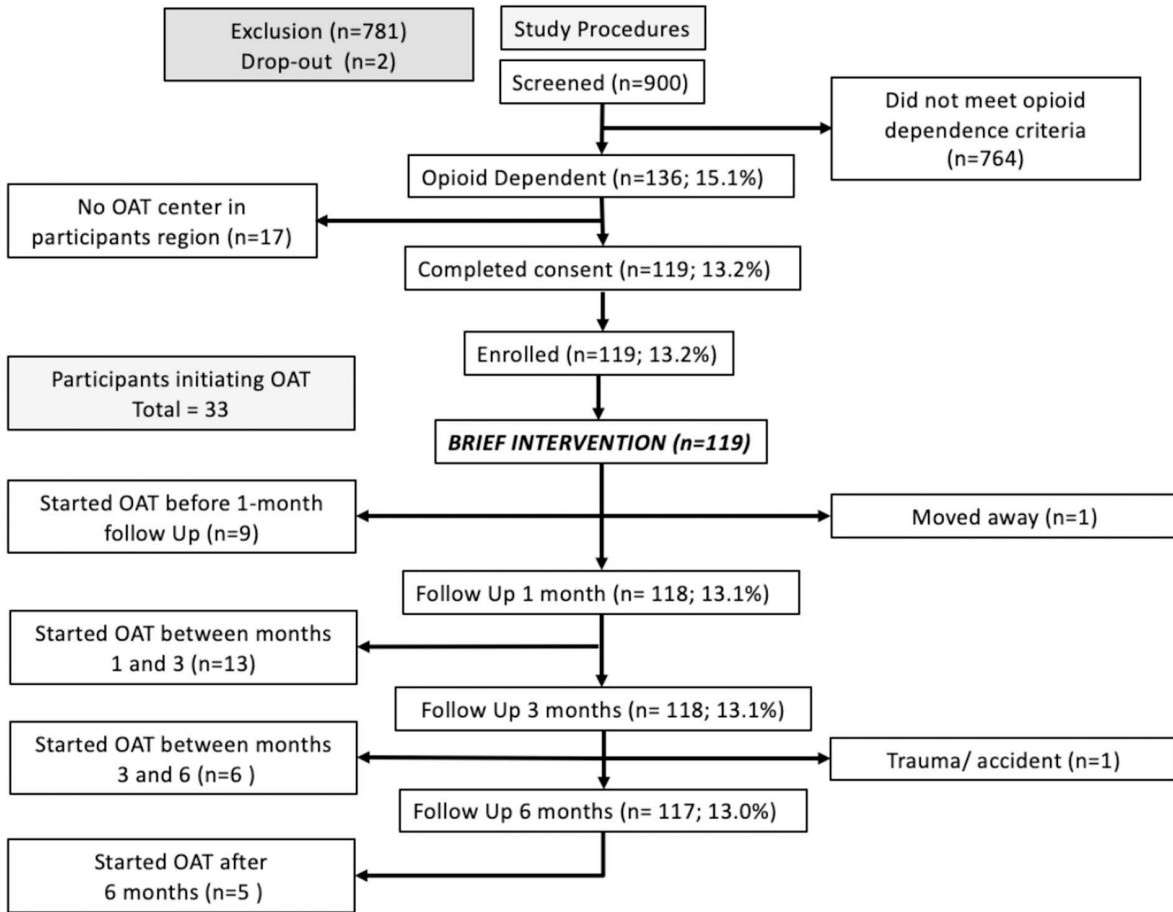

**Fig 1. Participant disposition.**

**Table 1. Baseline participant on probation with opioid use disorder, stratified by initiation of OAT (N = 119).**

| Characteristic | Total | Initiated OAT — Yes | Initiated OAT — No | p-value |
|---|---|---|---|---|
| | **N = 119 (%)** | **N = 33 (27.7%)** | **N = 86 (72.3%)** | |
| *Sites (N = 7)** | | | | <0.001 |
| A | 43 (36.1) | 10 (30.3) | 33 (38.4) | |
| B | 18 (15.1) | 7 (21.2) | 11 (12.8) | |
| C | 31 (26.1) | 2 (6.1) | 29 (33.7) | |
| D | 8 (6.7) | 3 (9.1) | 5 (5.8) | |
| E | 6 (5.0) | 1 (3.0) | 5 (5.8) | |
| F | 11 (9.2) | 8 (24.2) | 3 (3.5) | |
| G | 2 (1.7) | 2 (6.1) | 0 (0.0) | |
| *Sex* | | | | 0.018 |
| Male | 108 (91.5) | 27 (81.8) | 81 (95.3) | |
| Female | 10 (8.5) | 6 (18.2) | 4 (4.7) | |
| *Mean Age, years* | 36.4 (8.1) | 40.2 (8.1) | 34.9 (7.6) | 0.001 |
| *Partnered* | | | | 0.106 |
| Yes | 37 (31.4) | 14 (42.4) | 23 (26.7) | |
| No | 81 (68.6) | 19 (52.4) | 62 (72.9) | |
| *Children* | | | | 0.042 |
| Yes | 47 (39.8) | 18 (54.5) | 29 (33.7) | |
| No | 71 (60.2) | 15 (45.5) | 56 (65.1) | |
| *Education* | | | | 0.699 |
| Less than high school | 57 (48.3) | 15 (45.5) | 42 (49.4) | |
| Completed high school or higher | 61 (51.7) | 18 (54.5 | 43 (50.6) | |
| *Housing* | | | | 0.052 |
| Own residence (rent/own) | 25 (21.0) | 11 (33.3) | 14 (16.3) | |
| Friend or Relative's home | 88 (73.9) | 22 (66.7) | 66 (76.7) | |
| Homeless | 6 (5.0) | 0 | 6 (7.0) | |
| *Employment* | | | | 0.91 |
| Full or part-time employment | 72 (60.5) | 24 (72.7) | 48 (55.8) | |
| No employment | 47 (39.5) | 9 (27.3) | 38 (44.2) | |
| *Previously held in pre-trial detention center* | | | | 0.813 |
| Yes | 81 (68.1) | 23 (69.7) | 58 (67.4) | |
| *Previously held in prison (colony)(among those previously ever having been in a pre-trial detention center, N = 81)* | | | | 0.059 |
| Yes | 55 (67.9) | 17 (85.0%) | 38 (62.3) | |
| *Registered in the national narcology registry* | | | | 0.052 |
| Yes | 99 (83.2) | 31 (93.9) | 68 (79.1) | |
| *Previously attempted to receive OAT*** | | | | 0.126 |
| Yes | 22 (18.5) | 9 (27.3) | 13 (15.1) | |
| *Self-rated health status* | | | | 0.184 |
| Poor/fair | 87 (73.1) | 27 (81.8) | 60 (69.8) | |
| Greater than fair | 32 (26.9) | 6 (18.2) | 26 (30.2) | |
| **Screening for Medical Comorbidity** | | | | |
| *HIV positive* | | | | 0.027 |
| Yes | 27 (22.7) | 12 (36.4) | 15 (17.4) | |

*(Continued)*

**Table 1.** (Continued)

| Characteristic | | Initiated OAT | | | p-value |
|---|---|---|---|---|---|
| | Total | Yes | No | | |
| | N = 119 (%) | N = 33 (27.7%) | N = 86 (72.3%) | | |
| *HCV antibody status* | | | | | 0.006 |
| Positive | 66 (55.5) | 25 (75.8) | 41 (47.7) | | |
| Negative | 53 (44.5) | 8 (24.2) | 45 (52.3) | | |
| *Syphilis (RPR status)* | | | | | 0.006 |
| Positive | 17 (14.3) | 0 (0.0) | 17 (19.8) | | |
| Negative | 102 (85.7) | 33 (100.0) | 69 (80.2) | | |
| *Depression, moderate to severe* | | | | | 0.281 |
| Yes | 91 (76.5) | 23 (69.7) | 68 (79.1) | | |
| **Drug Use Characteristics** | | | | | |
| *Self-reported overdose (last 6 months)* | | | | | 0.204 |
| Yes | 47 (39.5) | 10 (30.3) | 37 (43.0) | | |
| *Injection frequency (last 30 days)* | | | | | 0.147 |
| Daily | 29 (24.4) | 5 (15.2) | 24 (27.9) | | |
| Less than daily | 90 (75.6) | 28 (84.8) | 62 (72.1) | | |
| *Drug addiction severity (DAST-10)* | | | | | 0.068 |
| Low/Moderate | 9 (7.7) | 0 (0.0) | 9 (10.7) | | |
| Substantial | 28 (23.9) | 6 (18.2) | 22 (26.2) | | |
| Severe | 80 (68.4) | 27 (81.8) | 53 (63.1) | | |
| **Social support** | | | | | |
| *Social support scale score* | 3.3 (1.0) | 3.5 (0.7) | 3.2 (1.1) | | 0.092 |
| *Receive social support from probation officers* | | | | | 0.568 |
| Yes | 10 (8.4) | 2 (6.1) | 8 (9.3) | | |
| *Want social support from probation officers* | | | | | 0.009 |
| Yes | 46 (38.7) | 19 (57.6) | 27 (31.4) | | |

Legend: OAT: opioid agonist treatment.

\* Ten sites agreed to participate, but only 7 screened and enrolled.

\*\* Includes participants who had previously sought treatment but did not initiate or complete OAT.

Note: Data are presented as mean (SD) for continuous variables or number and percentage for categorical variable.

## Participant characteristics

The mean age of enrolled participants was 36.4 (SD = 8.1) years, 91.5% were male, 55.5% had screening that was positive for HCV antibodies, 76.5% met screening criteria for moderate to severe depression (CES-D score ≥10), and 22.7% were HIV positive (7 of these or 25.9% were newly diagnosed with HIV). Forty-seven (39.5%) self-reported an overdose in the last 6 months. Those who started MOUD were significantly more likely to be older, female, have HIV or HCV infection, and to rate MOUD as an important treatment option compared with those who did not initiate MOUD."

## Treatment outcomes

ROC analysis demonstrated excellent discrimination of baseline MOUD interest for predicting MOUD initiation (AUC = 0.86). An interest score of ≥8 maximized sensitivity and specificity and was therefore selected as the threshold for defining high MOUD interest in subsequent analyses. Using the cutoff of ≥8, 23 (16.9%) stated they were interested in initiating MOUD before the brief intervention, which increased to 27 (19.9%) afterwards. Using a lower cutoff of ≥4, 103 (75.7%) expressed interest in

initiating MOUD and did not increase appreciably after the brief intervention (N = 104, 76.5%). Median interest (Table 2) in starting MOUD did not increase following the brief intervention (median [IQR] pre-intervention: 5 [3–7] vs post-intervention: 5 [3–7]; Wilcoxon signed-rank test, p<.001). Although the pre- and post-intervention summary medians were similar, paired analysis showed a statistically significant upward shift in MOUD interest scores after the brief intervention (Wilcoxon signed-rank test, p<0.001), indicating a small but consistent within-person increase. Median participants' attitudes towards and knowledge about treatment also remained unchanged after the brief intervention (Table 3). The perceived effectiveness of these treatments differed significantly between those who initiated MOUD and those who did not. Those who initiated MOUD perceived greater effectiveness of MOUD, family support, being in new environment, religion, detox (supervised withdrawal), inpatient rehabilitation, incarceration, employment, and other treatments with medication than those who did not initiate MOUD (p<0.05) (Table 4). Reasons for not initiating MOUD among those who did not initiate MOUD are presented in Table 5. Reasons for not being interested in MOUD initiation among PWID who have never been on MOUD are presented in Table 6.

Overall, 33 (27.7%) participants with OUD initiated MOUD. Nine started within 30 days after the brief intervention and 24 started after 30 days. Among participants who initiated MOUD, 72.7% had not previously attempted MOUD. In the multi-variable analysis, covariates included in the logistic regression model are age, sex, HIV status, HCV status, prior MOUD attempt, registration in the narcology registry, and post-intervention importance score for MOUD. Analysis showed a higher level (score ≥8) of perceived importance for MOUD as an effective treatment (post-intervention) for OUD, having HCV (aOR = 4.96, CI = 1.57-18.25), and having HIV (aOR 4.02, CI = 1.11-16.02,) predicted MOUD treatment entry (Table 7). Of the 33 that initiated MOUD, 32 (97%) remained on treatment through 6 months (Fig 2).

Of the 33 participated who started OAT during the study, 1 (3%) stopped OAT before their month 6 follow up interview. The remaining 32 participants were still on OAT at the time of their 6 month follow up.

**Table 2. Changes in interest in, difficulty in accessing, and importance to initiate medications for opioid use disorder before and after a brief intervention among participants on probation with opioid use disorder (N = 117).**

|  | Pre- Brief Intervention | Post-Brief Intervention | *p*-value |
|---|---|---|---|
| Interest in MOUD Score, median (interquartile range - IQR) | 5.0 (3.0 − 7.0) | 5 (3.0 − 7.0) | **<0.001** |
| Difficulty in Obtaining MOUD Score, median (IQR) | 5.0 (3.0 − 8.0) | 5.0 (2.0 − 7.0) | 0.67 |
| Importance of Starting MOUD Score, median (IQR) | 5.0 (3.0 − 8.0) | 5.0 (3.0 − 8.0) | 0.23 |

**Table 3. Comparison of mean MOUD attitude and knowledge scores* between baseline and 1 month among participants on probation with opioid use disorder (n = 118).**

| Statement[a] | Baseline Mean (SD) | Follow Up Mean (SD) | *p*-value |
|---|---|---|---|
| MOUD services should be available in the community so that all people who suffer from opioid addiction and OAT can receive it. | 3.86 (1.15) | 3.95 (1.06) | 0.67 |
| MOUD services should be introduced into prisons so that all inmates who suffer from opioid addiction and want opioid substitution therapy can receive it. | 3.35 (1.56) | 3.32 (1.52) | 0.85 |
| MOUD reduces opioid dependent individuals' risk of acquiring or transmitting HIV. | 3.83 (1.19) | 3.92 (1.11) | 0.67 |
| MOUD reduces addicts' criminal activities. | 3.45 (1.45) | 3.43 (1.33) | 0.74 |
| MOUD improves adherence to HIV medications in HIV-infected opioid dependent individuals. | 3.60 (1.30) | 3.73 (1.17) | 0.56 |
| MOUD reduces opioid dependent individuals' consumption of illicit opiates. | 3.39 (1.33) | 3.53 (1.18) | 0.51 |
| MOUD increases opioid dependent patients' adherence to tuberculosis medication | 3.44 (1.35) | 3.68 (1.19) | 0.23 |
| MOUD decreases opioid dependent individuals' risk of dying from overdose. | 3.85 (1.35) | 3.97 (1.24) | 0.59 |
| For people who have opioid addiction, it would be much better to treat them with buprenorphine rather than methadone. | 2.80 (1.19) | 2.83 (1.23) | 0.94 |

[a] Range of responses is 1–5 (1 = strongly disagree, 5 = strongly agree); MOUD refers only to methadone or buprenorphine as opioid agonist therapies.

**Table 4. Perceived treatment effectiveness of various treatment strategies among participants on probation with opioid use disorder.**

| How effective are each of these approaches in treating opioid dependence?[a] | | Started MOUD | | p-value |
|---|---|---|---|---|
| | Total | Yes | No | |
| | N = 119 | N = 33 | N = 86 | |
| | Mean (SD) | Mean (SD) | Mean (SD) | |
| Family support | 3.94 (1.37) | 4.45 (0.97) | 3.74 (1.45) | **0.001** |
| Move to a new environment | 3.76 (1.35) | 4.18 (1.07) | 3.59 (1.42) | **0.042** |
| Opioid agonist therapies | 3.22 (1.21) | 4.09 (0.91) | 2.88 (1.14) | **<0.001** |
| Inpatient rehabilitation | 3.21 (1.43) | 3.70 (1.16) | 3.02 (1.48) | **0.031** |
| Employment | 3.02 (1.56) | 3.61 (1.25) | 2.79 (1.61) | **0.015** |
| Detox | 3.0 (1.47) | 3.45 (1.30) | 2.83 (1.50) | **0.049** |
| Religion | 2.76 (1.49) | 3.48 (1.33) | 2.49 (1.47) | **0.001** |
| Treatment with medication (other than MOUD) | 2.53 (1.31) | 3.06 (1.0) | 2.33 (1.36) | **0.003** |
| Incarceration | 1.92 (1.28) | 2.36 (1.29) | 1.76 (1.25) | **0.004** |
| Alternative medicine | 1.56 (0.99) | 1.73 (0.98) | 1.5 (0.99) | 0.077 |

*Range of responses 1-5 (1= Not effective, 5 Very effective).

**Table 5. Reason for initiating MOUD among people who initiated MOUD.**

| Reasons | Total N = 32 (%) | Previously attempted OAT N = 9 | Never attempted OAT N = 23 |
|---|---|---|---|
| Wanted to improve my health | 30 (93.8) | 9 (100.0) | 21 (91.3) |
| Wanted to make a change in the circles I was moving in | 23 (71.9) | 9 (100.0) | 14 (60.9) |
| I was using too much and wanted to reduce my drug use | 22 (68.8) | 9 (100.0) | 13 (56.52) |
| I wanted to stop using illegal opioids permanently | 20 (62.5) | 9 (100.0) | 11 (47.8) |
| Financing drug consumption was too expensive | 20 (62.5) | 7 (77.8) | 13 (56.5) |
| Wanted to take better care of my family | 20 (62.5) | 3 (33.3) | 17 (73.9) |
| I needed a break from my habit because life was too chaotic | 14 (43.8) | 4 (44.4) | 10 (43.5) |
| I was afraid I might overdose | 10 (31.3) | 3 (33.3) | 7 (30.43) |
| Concern of being arrested/ imprisoned | 9 (28.1) | 2 (22.2) | 7 (30.4) |
| Wanted to stop committing crimes for my habit | 8 (25.0) | 3 (33.3) | 5 (21.7 |
| My family wanted me to start | 8 (25.0) | 1 (11.1) | 7 (30.4) |
| Afraid of getting an infection or contracting a disease | 6 (18.8) | 1 (11.1) | 5 (21.7) |
| Afraid of losing my job | 6 (18.8) | 0 (0.0) | 6 (26.1) |
| I want to be able to work again | 5 (15.6) | 1 (11.1) | 4 (17.4) |
| I wanted to get clean to get my HIV treated | 4 (12.5) | 3 (33.3) | 1 (4.4) |
| Other | 3 (9.4) | 0 (0.0) | 3 (9.4) |
| Another person wanted me to start | 2 (6.3) | 0 (0.0) | 2 (8.7) |
| Pregnancy | 0 (0.0) | 0 (0.0) | 0 (0.0) |

## Discussion

Consistent with implementation science frameworks, this study focuses on early feasibility indicators rather than down-stream implementation outcomes [69]. The study highlights the high prevalence of OUD (15.1%) and HIV (22.7%) among individuals on probation in Moldova, emphasizing the critical role of probation settings as a touchpoint to identify people eligible for treatment (and prevention). By employing a modified SBIRT (SET: screen-evaluate-treat) strategy, the intervention

**Table 6. Reasons for not initiating OAT among those declining OAT (N = 86).**

| Reasons | N (%) |
|---|---|
| My friends had a bad experience on OAT | 13 (15.1) |
| I think methadone is just another drug | 10 (11.6) |
| I'll have to use methadone my whole life | 8 (9.3) |
| Methadone is bad for my health | 7 (8.1) |
| Did not want to get registered | 6 (7) |
| My friends and family have negative feelings about OAT | 3 (3.5) |
| Methadone is free and therefore bad quality | 2 (2.3) |
| I don't want my friends to know I'm on treatment | 1 (1.2) |

**Table 7. Predictors of initiating OAT among people with opioid use disorder (N = 119).**

| Variable | OR (95% CI) | aOR (95% CI) |
|---|---|---|
| Importance of OAT for treating opioid use disorder (post-intervention, score ≥8)** | 1.85 (1.49-2.40, p < 0.001) | 1.93 (1.53-2.57, p < 0.001) |
| Previous OAT entry attempts | 2.11 (0.78-5.52, p = 0.131) | |
| Previously registered at narcology addiction treatment center | 4.10 (1.09-26.82, p = 0.069) | 3.17 (0.54-28.30, p = 0.239) |
| HIV positive | 2.70 (1.09-6.70, p = 0.031) | 4.02 (1.11-16.02, p = 0.038) |
| HCV antibody positive | 3.43 (1.44-8.91, p = 0.007) | 4.96 (1.57-18.25, p = 0.010) |

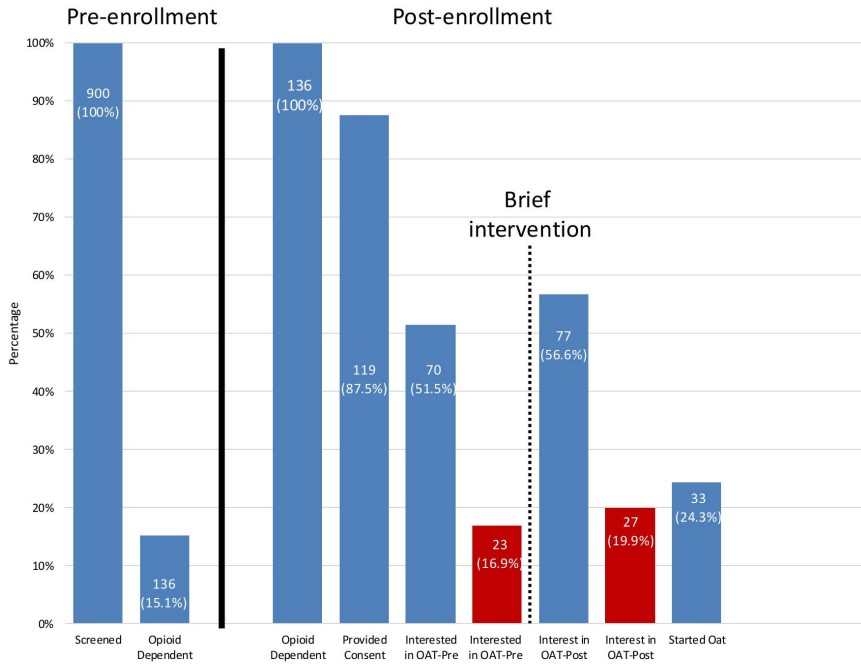

Legend: OAT = Opioid agonist therapy. BI = Brief intervention. Blue bars indicate pre- and post-interest using 4 or greater as a cut-off; red bars indicate using ≥8 or greater as cut-off.

**Fig 2. Opioid use disorder continuum of care cascade for people in probation in Moldova.**

successfully identified a high prevalence of individuals with OUD and HIV as well as interest in initiating MOUD with methadone or buprenorphine. This modified SBIRT strategy was effective at identifying individuals with OUD in probation settings and increasing their interest in MOUD. A proportion of participants subsequently initiated MOUD following the intervention, suggesting potential to improve treatment engagement, which warrants evaluation in controlled studies.

Key among the findings is that over a quarter of the 119 participants with OUD initiated or remained on MOUD, with 97% being retained on treatment after six months. Additionally, the integrated screening identified substantial undiagnosed HIV cases, further underscoring the potential for probation to serve as an impactful entry point for syndemic screening and linking individuals to comprehensive treatment for chronic conditions. In the case of HIV, undiagnosed PWH substantially contribute to HIV transmission [70]. These findings, however, suggest that aligning probation's public safety mandate with public health goals will require structural and procedural adaptations so that the SBIRT can be implemented by probation staff rather than trained researchers.

Although the median interest in MOUD score and interquartile range were unchanged from pre- to post- brief intervention assessments (median 5 [IQR 3–7] at both time points), the post-intervention scores were shifted toward higher values overall (Wilcoxon signed-rank test, $p < 0.001$) after the brief intervention and 33 of 119 participants initiated MOUD and 32 were retained on treatment over 6-months. Predictors of initiating MOUD were perceiving MOUD to be an important treatment, having HCV and having HIV. Twenty-seven participants tested positive for HIV, with seven being newly diagnosed during screening procedures. Particularly in LMICs like Moldova where MOUD coverage is low and decarceration efforts have yet to be fully realized, probation has been an under-utilized opportunity to intervene with PWID to support access to prevention and treatment services. Interventions that target the probation period among PWID can be especially impactful as they can address both OUD and HIV among criminal legal-involved individuals. Notably, nearly 40% of participants reported experiencing an overdose within the past six months, a critical indicator of elevated risk for morbidity and mortality within this supervised population. Research indicates that individuals under probation face overdose rates **up to 15 times higher** than the general population, and overdose fatalities remain strikingly high during periods of community supervision [11,71]. This alarming prevalence underscores the urgent need for enhanced surveillance of OUD and rapid linkage to OAT like methadone, buprenorphine, [72] which are proven to substantially reduce overdose mortality [73] and help reduce the transmission of blood-borne infections like HIV and HCV [2–4]. This finding further highlights an important opportunity for probation officers to play a pivotal role in harm reduction efforts through proactive screening for substance use, facilitating access to OAT, and distributing naloxone to protect both individuals and the wider community. Findings here suggest that probation can and should serve as a key entry point into OUD treatment for PWID, which is similar to findings in nearby Ukraine [43] and points to the need for more broadly intervening in probation.

To reduce the burden on prisons, which have a mandate to provide healthcare that is equivalent to that in the community, the criminal legal system has transitioned many people who would otherwise be in prison to probation. A key tenet of probation, however, is to ensure that crime is reduced (i.e., either illicit drug use and/or the crimes committed to procure drugs). As MOUD is associated with substantial reductions in criminal activity, screening for OUD and supporting clients to gain treatment could be a critical strategy to align public safety and public health. As probation services are mostly designed to promote public safety, they have not yet has been leveraged to simultaneously promote public health. For probation to fully align their public safety mission with public health, it will be crucial to be able to implement SBIRT in such settings by the staff who operate probation as our findings were observed when people external to probation implemented the screening. Though not explicitly understood by probation officers, supporting their clients in accessing MOUD (a public health benefit) also promotes public safety by reducing or eliminating substance use, reducing other crime and supports overall recovery [7,74]. In real-world settings, developing strategies that allow probation officers to modify SBIRT procedures requires a clear understanding of how MOUD does promote public safety (through its reduced drugs use and criminal activity). Inquiring about substance use by probation officers, however, may require an alternative delivery strategy, perhaps by reducing opportunities to stigmatize and discriminate against those using drugs who are "breaking the law".

Redesigning or modifying SBIRT in probation might include deployment behavioral design interventions (BDI), which uses choice architecture, framing and nudging. Choice architecture focuses on the different way choices can be presented and how this presentation may impact decision-making [75]. For example, do clients self-screen by having a self-screening instrument (e.g., using a QR code, pen and paper or online decision tool) that is anonymous and provides an action-able output that asks probation officers to assist in supporting their recovery (without directly discussing illicit opioid use). Framing MOUD may also be impactful in whether people with OUD decide to initiate MOUD [76]. For example, it can be framed as a tool to manage one's addiction or as a strategy to remain in the community and not go to prison. Other types of framing may derive from the officer directly by stating upfront that their goal is to keep their clients in the community and one way of doing this for clients to engage in treatments that support their recovery. Framing can also be supported by other implementation tools like decision aids that allow patients to make preferences about treatment that are aligned with their values [77,78]. Last, nudging is a small push without forcing people to make decisions [79]. People with OUD may be "nudged" to make a healthier decision by including facts about rates of recidivism and contraction of infectious diseases among those who continue to inject opioids. Alternatively, probation officers themselves might be nudged to reduce their likelihood of sending people to prison by setting rewards for successful retention in community treatment. The aforementioned strategies taken together may be synergistic and promote the uptake of OUD treatment if implemented correctly.

Important in this pilot study was the integration of screening for comorbid conditions like OUD, HIV and HCV, which are syndemic [44] and well-suited to advance multiple public health goals [42]. These integrated programs could be even more impactful in high-prevalence settings or among key populations such as PWID in Moldova, where it is estimated that a third of all people in prisons use drugs and only 12% are prescribed MOUD [6]. MOUD remains one of the most effective and cost-effective strategies to prevent and treat HIV [23,80], thus making it an intervention to treat OUD and address the HIV epidemic. The secondary outcome of HIV identification was therefore a critical element of this intervention. Evidently, there remains a need to improve the uptake of MOUD among PWID and improve HIV case detection.

While 27.7% of our sample initiated MOUD as part of the brief intervention, the brief intervention component only modestly increased interest in starting MOUD and unfortunately, did not change attitudes towards or knowledge about MOUD. Although evidence for the effectiveness of brief interventions in reducing drug use is mixed, they do appear to have benefits within SBIRT protocols for alcohol and OUD. In this study, the findings align with studies showing modest improvements in motivation but limited effects on behavioral outcomes [81]. This may be due to persistent negative attitudes and misinformation about MOUD within prisons in Moldova [82]. Similar attitudes and beliefs have been found in other countries in EECA, including Kyrgyzstan [83] and Ukraine [56,41,84], and in the case of Kyrgyzstan, did not increase methadone scale-up [28,85]. More broadly across EECA, punitive "Russian narcology" legacies, reinforced by institutional surveillance and abstinence-oriented norms and compounded in many settings by weakened support for evidence-based harm reduction, can entrench stigma and misinformation about MOUD in ways that blunt the impact of brief, individually focused interventions [86]. One potential explanation is that the brief intervention was not effective because of content, while another explanation might be delivery where deliverers had their own implicit biases or presented findings not congruent with context or not aligned with patient preferences [87,88]. One alternative delivery strategy could be using either informed- or shared-decision aides which improve knowledge, reduce decisional conflict and stimulate individuals to be more active in decisions regarding their health [89]. Potential future options regarding choice architecture in BDIs may include embedding a modified SBIRT process initially within an informed (or shared) decision aid for MOUD and the referral for treatment could derive from the client providing a decision-aid output providing recommendations for the client.

Numerous studies have examined the unique social dynamics in prison settings that may dissuade engagement with OUD services [82,84,90]. The extent to which they influence initiating MOUD in probation, however, is not known. In some cases, criminal subcultures, or informal systems of governance, in prisons forbid the use of drugs, and understandings of methadone vary. Whether a particular prison's subculture deems methadone a drug or not can influence the uptake of methadone. Criminal subculture is deeply embedded in individuals leaving prisons as a social and cultural force which

may supersede any individual-level intervention like an SBIRT, hence the lack of increase in MOUD post-intervention [28]. Additionally, because probation in community settings can be an extremely stressful period for individuals with low social support, this may suggest the need to supplement support systems that drive subcultural norms, including social, physical and economic levels [91]. This is especially true for those who are never detained or who transition from pre-trial detention for their probation period. While little is known about how dynamics brought on by criminal subculture persist outside of the prison setting and into the probationary period, we can hypothesize that criminal subcultural influence is lower in probation than in prisons, but we do know whether these norms continue into the community and may impact probation. One study from Kyrgyzstan documented low levels of MOUD uptake in prison using SBIRT [28] while a similar study from Moldova showed that one quarter of those screened in prison with OUD initiated methadone, but most did so soon after release when the criminal subculture may have had less influence on daily activities [92]. Sociologically rigorous qualitative work is, therefore, necessary to elucidate the underlying social, structural and cultural forces that shape decision making about MOUD uptake outside the prisons setting. Regardless, initiation of MOUD was relatively high among PWID in this study.

Our findings, in part, are similar to findings in probation in Ukraine using a similar methodology where 24.6% of participants initiated MOUD, closely mirroring the 27.7% who did so in Moldova. Our findings differ, however, in the greater differences in perceived effectiveness of various treatments between those who did and did not initiate MOUD. Both studies found a modest but significant increase in interest in starting MOUD after the brief intervention [43]. In contrast to both these studies that demonstrated the utility of SBIRT for improving MOUD initiation, one systematic review and one meta-analysis found that brief screening interventions are unimpactful in improving the uptake of substance use treatment [93,94]; however, treatment services for OUD were poorly represented in these studies, and neither included interventions delivered in the probational setting [95], thus limiting the generalizability of their findings to our study setting and context. Given the little change in interest in MOUD after the intervention, modifying SBIRT to eliminate the brief intervention and perhaps creating a "green light" pathway to treatment for those interested might be considered.

A critical aspect of this intervention is the bundling of screening for comorbid conditions that occurred here, as 22.7% were found to be PWH, which was higher than the reported 11% of PWID with HIV in Moldova [96]. This may suggest that people on probation who also meet criteria for OUD are at a particularly high risk for HIV. Although this study was not primarily designed to identify new HIV cases, 25.9% of people with HIV in this study were newly identified and referred to care, an important tool in meeting UNAIDS 95-95-95 goals to reduce HIV incidence. If one extrapolates to the 11,970 on probation this would minimally translate to 1807 people identified with OUD, of which 486 could be initiated and retained on methadone and 106 new HIV infections identified.

Given nearly one third of people in Moldova do not know they have HIV, modified SBIRT screening that bundles screening for commonly co-occurring conditions like OUD, HIV and HCV can amplify the impact of such strategies [42,44]. Future studies should consider other medical conditions as a part of SBIRT screening including depression and other psychiatric disorders that often disproportionately afflict those with substance use disorders involved in criminal legal settings [97].

Despite the feasibility and important findings from our study, it does have limitations. First, this study did not assess implementation feasibility from the perspective of probation staff, including adoption, fidelity, or sustainability of SET delivered by probation officers themselves. These outcomes will be essential for future implementation trials. Second, our sample size of 119 is small and limits generalizability. This is, however, mitigated by screening a substantially larger number of 900 people on probation. Additionally, 17 of our screened participants were ineligible due to distance from a participating MOUD center. This exclusion criteria could introduce some bias as it likely would have been more difficult for those living further away from an MOUD site to initiate MOUD and be retained in treatment. Third, social desirability bias may have also influenced our findings. Although participants self-administered the survey on REDCap, there may have been social desirability bias introduced as they may have wanted to please motivational interviewers (research assistants). COVID-19 did not disrupt study procedures although it lengthened the recruitment phase. Fourth, though not included in our study was the

influence of the national registry for people with OUD (i.e., restrictions on employment and driving and acts as a deterrent to MOUD initiation), it may partially explain modest uptake despite interest. This may, in part, have explained why a larger number of people did not initiate MOUD. The pre/post design without a control group limits causal inference regarding intervention effects on MOUD interest and initiation compared with randomized or controlled designs. Finally, it is likely that some individuals with OUD did not want to report their drug use for fear it would violate the rules of their probation. OUD prevalence, therefore, was likely significantly higher among the 900 we screened than we were able to capture. It is important to note, however, among those who did disclose their drug use, there were no instances or complaints of breaches of confidentiality.

Notwithstanding these limitations, this modified SBIRT strategy deployed in probation by trained interventionists was effective in identifying individuals with OUD and increasing their interest in MOUD. Among participants with OUD who were not receiving MOUD at enrollment, a substantial subset initiated MOUD during follow-up after exposure to the intervention. Given there are greater numbers of people on probation than in prisons, the probation period could be an optimal time point to encourage PWID with OUD to initiate MOUD. Further research should explore other brief, rapidly introduced interventions that can address specific barriers to MOUD uptake, and qualitative studies are needed to explain why the brief intervention does not change attitudes nor interest – perhaps other decision support tools might. Policy reform may also be necessary so that people on probation can disclose their drug use without violating their probation.

## Conclusion

In Moldova, the prevalence of OUD and HIV is high among people on probation. The criminal legal system as it shifts toward decarceration will increasingly place more individuals who interface with probation, which is not linked to rights about healthcare, but could be reenvisioned to screen for and link individuals to treatment, especially ones like MOUD that are associated with reductions in crime (i.e., public safety). Rapid screening strategies can identify and provide opportunities for treatment for multiple chronic conditions, including HIV case detection. Implementation opportunities, however, will require aligning public safety mandates in probation with public health. Training probation officers in SBIRT may be an effective way to do so.

## Author contributions

**Conceptualization:** Lyu Azbel, Frederick L Altice.

**Data curation:** Matthew N Ponticiello, Daniel J Bromberg, Lyu Azbel, Sergiu Cugut, Svetlana Doltu, Frederick L Altice.

**Formal analysis:** Matthew N Ponticiello, Daniel J Bromberg.

**Funding acquisition:** Frederick L Altice.

**Investigation:** Daniel J Bromberg, Lyu Azbel, Svetlana Doltu, Frederick L Altice.

**Methodology:** Lyu Azbel, Svetlana Doltu, Frederick L Altice.

**Project administration:** Daniel J Bromberg, Lyu Azbel, Sergiu Cugut, Svetlana Doltu, Frederick L Altice.

**Resources:** Frederick L Altice.

**Supervision:** Lyu Azbel, Sergiu Cugut, Svetlana Doltu, Frederick L Altice.

**Validation:** Svetlana Doltu, Frederick L Altice.

**Visualization:** Matthew N Ponticiello.

**Writing – original draft:** Matthew N Ponticiello, Frederick L Altice.

**Writing – review & editing:** Matthew N Ponticiello, Daniel J Bromberg, Lyu Azbel, Svetlana Doltu, Frederick L Altice.

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
