## [Decision Letter · Decision Letter 0]

22 Apr 2025

PGPH-D-24-02730

Probation as a targeted entry point for scaling up opioid agonist therapies in Moldova using a modified screening, brief intervention and referral to treatment strategy

Dear Dr. Ponticiello,

Thank you for submitting your manuscript to PLOS Global Public Health. After careful consideration, we feel that it has merit but does not fully meet PLOS Global Public Health’s publication criteria as it currently stands. Therefore, we invite you to submit a revised version of the manuscript that addresses the points raised during the review process.

Please note that we have only been able to secure a single reviewer to assess your manuscript. We are issuing a decision on your manuscript at this point to prevent further delays in the evaluation of your manuscript. Please be aware that the editor who handles your revised manuscript might find it necessary to invite additional reviewers to assess this work once the revised manuscript is submitted. However, we will aim to proceed on the basis of this single review if possible. The reviewer has raised a number of concerns that need attention. Could you please revise the manuscript to carefully address the concerns raised?

We look forward to receiving your revised manuscript.

Kind regards,

Johanna Pruller, Ph.D.

PLOS Staff Editor

Journal Requirements:

1. Please provide additional information regarding the considerations made for the participants included in this study. For instance, please discuss whether participants were able to opt out of the study and whether individuals who did not participate receive the same treatment offered to participants. 2. Please include a complete copy of PLOS’ questionnaire on inclusivity in global research in your revised manuscript. Our policy for research in this area aims to improve transparency in the reporting of research performed outside of researchers’ own country or community. The policy applies to researchers who have travelled to a different country to conduct research, research with Indigenous populations or their lands, and research on cultural artefacts. The questionnaire can also be requested at the journal’s discretion for any other submissions, even if these conditions are not met. Please find more information on the policy and a link to download a blank copy of the questionnaire here: https://journals.plos.org/globalpublichealth/s/best-practices-in-research-reporting. Please upload a completed version of your questionnaire as Supporting Information when you resubmit your manuscript. 3. Your current Financial Disclosure states, “The authors have no financial conflicts of interest to disclose.”. However, your funding information on the submission form indicates that you received funding from “National Institute for Drug Abuse and Foundation for the National Institutes of Health with grant numbers F31DA054861, T32 GM13665 and T32MH020031”. Please indicate by return email the full and correct funding information for your study and confirm the order in which funding contributions should appear. Please be sure to indicate whether the funders played any role in the study design, data collection and analysis, decision to publish, or preparation of the manuscript. 4. Please provide separate figure files in .tif or .eps format. For more information about figure files please see our guidelines:  https://journals.plos.org/globalpublichealth/s/figures https://journals.plos.org/globalpublichealth/s/figures#loc-file-requirements 5. In the online submission form, you indicated that Data will be made available upon reasonable request.  All PLOS journals now require all data underlying the findings described in their manuscript to be freely available to other researchers, either 1. In a public repository, 2. Within the manuscript itself, or 3. Uploaded as supplementary information. This policy applies to all data except where public deposition would breach compliance with the protocol approved by your research ethics board. If your data cannot be made publicly available for ethical or legal reasons (e.g., public availability would compromise patient privacy), please explain your reasons by return email and your exemption request will be escalated to the editor for approval. Your exemption request will be handled independently and will not hold up the peer review process, but will need to be resolved should your manuscript be accepted for publication. One of the Editorial team will then be in touch if there are any issues.

Additional Editor Comments (if provided):

Reviewers' comments:

Reviewer's Responses to Questions

**Comments to the Author**

1. Does this manuscript meet PLOS Global Public Health’s publication criteria? Is the manuscript technically sound, and do the data support the conclusions? The manuscript must describe methodologically and ethically rigorous research with conclusions that are appropriately drawn based on the data presented.

Reviewer #1: Partly

2. Has the statistical analysis been performed appropriately and rigorously?

Reviewer #1: Yes

3. Have the authors made all data underlying the findings in their manuscript fully available (please refer to the Data Availability Statement at the start of the manuscript PDF file)?

Reviewer #1: No

4. Is the manuscript presented in an intelligible fashion and written in standard English?

Reviewer #1: Yes

5. Review Comments to the Author

Reviewer #1: The submitted manuscript addresses a relevant topic and an important opportunity to scale-up OAT programs in Moldova through the probation system. Also, this article reflects the results of one of two studies conducted using the same design in parallel, the second study was conducted in Ukraine. Both countries have similar historical backgrounds and, as a result, similar legal and prison system problems and characteristics of the HIV epidemic, which has been concentrated among people who inject drugs for decades. Therefore, conducting such studies and publishing articles based on their findings is crucial. However, despite the many systemic similarities between Ukraine and Moldova, there are also significant differences, particularly in the implementation of the probation system, which was established in Moldova much earlier than in Ukraine. I focus on research in Ukraine, specifically in prisons, but I am also familiar with the prison contexts of the surrounding countries. In particular, I know that until very recently, the main probation efforts in Moldova focused on penitentiary probation, which represented a series of measures taken by prison officers shortly before the release of people serving prison sentences. In Ukraine, on the contrary, significant efforts were directed at establishing probation as an alternative to imprisonment. The article does not explain what type of probation is being discussed or how it is organized at all, and individual phrases that do not constitute a general context only confuse understanding. As it stands now, the text itself is substantively lacking in contextual data, and more broadly–given that the two studies were conducted in two countries in parallel using the same protocol–in methodological consistency. For example, the Study setting subsection states that

The probation system combines 38 probation offices(41) to oversee persons deemed stable enough to be supervised in community settings. It is linked to social rehabilitation centers that supplement these services in three territorial regions: north, center and south. At the end of 2018, there were 11,970 people on probation(42). (147-151, p. 6)

Here it is necessary to clarify what these social rehabilitation centers are, since you mention them, and how they differ from prisons, and what civil status do the people who stay in them or systematically visit them have? And what is the connection between these 38 offices and the three social rehabilitation centers? And under what conditions can people get into probation, if it is an alternative to imprisonment (e.g. for minor offenses, first encounter with the criminal system, etc.).

Further in the Results section, in Table 1. Baseline participant characteristics, stratified by initiation of OAT (N = 119), one of the participant characteristics is Previously held in pre-trial detention center, but what does this characteristic tell us if we do not understand the conditions of probation entry and what type of probation we are talking about? In any case, it seems strange to ask only whether the participant has been Previously held in pre-trial detention center (SIZO) without asking whether he/she has been previously held in prison(colony), unless the mere possibility of being placed on probation excludes the possibility of having previous experience of imprisonment in a colony. In any case, this question should be clarified.

Finally, in the Discussion section, the authors point out that

probation is an extremely stressful period for individuals with low social support. This may suggest the need to create structural interventions that pay attention to what drives subcultural norms, including social, physical and economic levels(66) in order to address them before people in prison are moved to the probation period. (425-429, p. 22)

Due to the lack of information about the conditions of probation, the first statement looks too strong and in no way linked to the data discussed in the article. In the next sentence, it says that ‘in order to address them before people in prison are sentenced to the probation period’, which means that the authors were talking about penitentiary probation, i.e. prison-based probation? Why then do the authors go on to write that

While little is known about how dynamics brought on by criminal subculture persist outside of the prison setting and into the probation period, we can hypothesize that criminal subcultural influence is lower in probation than in prisons, but we do know that these norms continue into the community and may impact probation’ (429-432, p. 22).

Although here I would like to point out that this phrase once again confuses the understanding of how the probation system is implemented in Moldova, I would also like to ask why attitudes learned while serving prison time (ponyatiya) cannot be transferred into life after people are released? Does Moldova, as in other post-Soviet countries, have an effective system of social re-integration of former prisoners, which allows people not to return to similar communities outside the prison walls?

Although I have no questions about the reliability of the findings, I would like to draw the authors' attention to the internal contextualization of the study results, which opens up new possibilities for interpreting the data. In particular, this may help to better understand some of the issues that the authors raise in the discussion, such as the low motivation of participants to engage in OAT or the persistence of negative attitudes about OAT. For example, in the Results section, Table 1. Baseline participant characteristics, stratified by initiation of OAT (N = 119), among participant characteristics is the parameter ‘Registered at narcology addiction treatment center’. It is further mentioned in Table 6: Reasons for not initiating OAT among people who did not initiate OAT (N = 86). Out of 86 people who did not initiate OAT, only six indicated that they ‘Did not want to get registered’. If we go back to Table 1, to the ‘Registered at narcology addiction treatment center’ characteristic, we can notice that out of 86 people who did not initiate OAT, only 18 responded that they were not ‘Registered at narcology addiction treatment center’. Therefore, considering ‘'Did not want to get registered’ as a reason for not initiating OAT can only be relevant for the 18 people who were not registered before. Whereas 6 out of 18 is a very different proportion and therefore a very different position for this reason in the overall hierarchy of reasons for not initiating OAT treatment. I would recommend the authors to pay special attention to the topic of mandatory registration of people with OUD to be eligible for OAT. A number of studies in Ukraine have addressed this issue, where the emergence of private OAT programs, which do not always require confirmation of OUD diagnosis, automatically removes the problem of mandatory registration if people have the money to pay for the services of a private OAT program.

I would like to draw the attention of the authors to the use of the concept ‘criminolegal’, in particular in the case of ‘criminolegal settings’ in abstract and several times in the text and ‘From a criminolegal perspective’ in Introduction, p. 3, 63. If there is some sense in using this particular concept, which most resembles the calque from Russian (ugolovno-pravovoy), which may have diffused into the normative documents of other Soviet countries during the Soviet era, then it is worth explaining its use. If not, it is worth replacing it with a more widely accepted one.

One last minor point. The Results section begins with a subsection on Participant disposition, stating that ‘Forty-sever (39.5%) overdosed in the last 6 months’. The authors do not revisit this result again, then why is it important to mention here? While it also seems important to me, I would like to see the authors' comments in the discussion.

I would like to end by saying that, as I wrote at the beginning, the publication of this article is both important and needed, and I hope that my comments will help to make this text even better.

6. PLOS authors have the option to publish the peer review history of their article (what does this mean?). If published, this will include your full peer review and any attached files.

**Do you want your identity to be public for this peer review?** For information about this choice, including consent withdrawal, please see our Privacy Policy.

Reviewer #1: No

---

## [Decision Letter · Decision Letter 1]

29 Oct 2025

PGPH-D-24-02730R1

Probation as a targeted entry point for scaling up opioid agonist therapies in Moldova using a modified screening, brief intervention and referral to treatment strategy

Dear Dr. Ponticiello,

Thank you for submitting your manuscript to PLOS Global Public Health. After careful consideration, we feel that it has merit but does not fully meet PLOS Global Public Health’s publication criteria as it currently stands. Therefore, we invite you to submit a revised version of the manuscript that addresses the points raised during the review process.

The manuscript has been evaluated by two reviewers, and their comments are available below.

The reviewers have raised a number of concerns that need attention. Could you please revise the manuscript to carefully address the concerns raised?

We look forward to receiving your revised manuscript.

Kind regards,

Johanna Pruller, Ph.D.

PLOS Staff Editor

Journal Requirements:

Reviewers' comments:

Reviewer's Responses to Questions

**Comments to the Author**

1. If the authors have adequately addressed your comments raised in a previous round of review and you feel that this manuscript is now acceptable for publication, you may indicate that here to bypass the “Comments to the Author” section, enter your conflict of interest statement in the “Confidential to Editor” section, and submit your "Accept" recommendation.

Reviewer #1: All comments have been addressed

Reviewer #2: (No Response)

2. Does this manuscript meet PLOS Global Public Health’s publication criteria? Is the manuscript technically sound, and do the data support the conclusions? The manuscript must describe methodologically and ethically rigorous research with conclusions that are appropriately drawn based on the data presented.

Reviewer #1: Yes

Reviewer #2: Partly

3. Has the statistical analysis been performed appropriately and rigorously?

Reviewer #1: Yes

Reviewer #2: No

4. Have the authors made all data underlying the findings in their manuscript fully available (please refer to the Data Availability Statement at the start of the manuscript PDF file)?

Reviewer #1: Yes

Reviewer #2: No

5. Is the manuscript presented in an intelligible fashion and written in standard English?

Reviewer #1: Yes

Reviewer #2: Yes

6. Review Comments to the Author

Reviewer #1: (No Response)

Reviewer #2: Thank you for the opportunity to review the manuscript entitled ‘Probation as a targeted entry point for scaling up opioid agonist therapies in Moldova using a modified screening, brief intervention and referral to treatment strategy.’ The authors estimate OUD, HIV, and HCV prevalence, measure MOUD uptake and retention, and evaluate the impact of a brief motivational interviewing intervention on MOUD interest among people on probation with opioid use disorder. While the study is important and interesting, the presentation of the study – especially the objectives, methods and results – is lacking in clarity and organization. Specific comments are below.

ABSTRACT

-In Methods or Results, would be helpful to very briefly detail the ‘MOUD interest score’ – for example, is this a validated instrument or 1 question and what is the possible score range and directionality?

-Just a note that any changes below should be integrated into the Abstract, where appropriate.

INTRODUCTION

-Would add that another important individual and public health benefit of MOUD is decreased risk of overdose.

-Suggest adding a bit more information on the typical MOUD programs available in Moldova – methadone predominantly? Free-of-charge from the public sector? In hospitals/clinics or community-based locations?

-For those unfamiliar with the legal system, would be good to clarify whether those going into probation are coming out from prison or straight from sentencing – or a mix. Or maybe that is what ‘post-released from criminal sentences’ means – suggest revision to clarify.

-The authors mention that one of primary goals of the study was to pilot test the feasibility of targeting probation as an opportunity to encourage MOUD, but it isn’t clear after reading the paper how ‘feasibility’ is defined or assessed. Suggest the authors more clearly define what their aims are for this paper.

METHODS

-Much of the extensive background on probation in Moldova seems better suited for the Introduction than Methods and there is likely more detail than required (e.g., probably don’t need to know the Law No. that governs probation, the required education for probation officers) – but defer to editor on these.

-This sentence is not clear as currently written: “To avoid potential conflict between officer and client, the research team was allowed to be present in the office where the officer typically meets with clients every 2-4 weeks.” How does this avoid conflict between the client and officer?

-Related to the daily caseloads of probation officers – these seem really high but maybe the interpretation of ‘daily’ could be clarified as it could be interpreted as the number of people the officers see/interact with each day.

-Participants – Would add date range in which clients were approached (assuming enrollment was not the full study period [01/11/2019 to 30/04/2023]?)

-Participants – If space allows – would suggest the authors include the full text of the ‘single item screening question’ for opioid use. And, is this just for opioid use or opioid use disorder?

- Participants – The authors detail the confirmation of opioid use disorder in the next section but would very briefly mention this in the ‘Participants’ section. Maybe within the inclusion criteria – e.g., ‘OUD diagnosis using the brief Rapid Opioid Dependence Screen’ – if this interpretation of screening and inclusion is correct.

-Suggest removing ‘Study procedures are similar as those described in Ukraine(40) as this was part of a two-site study’, as this has already been mentioned.

-‘Both methadone and buprenorphine were offered as treatment’ (line 291-292) – this seems oddly placed and sounds as if the research assistants were offering treatment in the probation office. Suggest that authors revise.

-‘In this study, the only MOUD available are opioid agonist therapies (OAT) including methadone or buprenorphine… Moldova’s Ministry of Health provides MOUD for free in prisons and within the community, including methadone and buprenorphine…’ – these sentences are oddly placed (better in Introduction, probably) and confusing as written. Did the study provide MOUD or are the authors referring to the publicly available MOUD?

-Were those currently uninterested in MOUD at least given referral information? If not, why?

-‘Participants could enroll in MOUD up to 6 months after their brief intervention’ (lines 304-305). It is unclear why clients would have MOUD access restricted at any point. Maybe the authors meant that they assessed MOUD initiation up to 6 months after some time point (specific time point being X)? Relatedly, for MOUD retention, participants were followed for 6 months following what time point (study enrollment or MOUD initiation?). Suggest authors revise for clarity.

-If space allows, it would be helpful to understand what exact data was extracted from the MOUD clinical sites (medication prescribed, refills dates, etc.) and how study staff were able to link study participants to patient records (e.g., unique ID, name).

-‘but not beyond that’ (lines 308-309) is not needed and suggest removing.

-More details on the lab study procedures is needed – where was the rapid testing and counseling completed? The probation office? Was the HCV testing antibody only or also RNA? What was the syphilis testing exactly – RPPR and/or TPHA?

-There is an insufficient amount of information provided on what exactly the intervention entailed besides that it was motivational interviewing. This section needs considerable attention and revision so that readers can put findings in context. Also, for the sake of clarity, suggest that the details on the intervention (i.e., motivational interviewing) be its own section separate from data collection procedures.

-The scale mentioned for difficulty in accessing MOUD and importance of starting MOUD seems not to match – ‘0 indicated no interest at all and 10 indicated the highest level of interest possible.’ This appears to just be the scale for interest. Suggest authors revise.

-For what is assumed to be the primary outcome of interest – the ‘MOUD interest score’ – the authors mention that it was measured ‘before and after the brief intervention’ – does ‘after’ mean immediately after the baseline intervention session (same day as the baseline visit) or after the second session which was 30 days after the initial intervention. This timeframe is very unclear as written. Additionally, has this ‘score’ been validated?

-Line 323 – the word ‘standardized’ is repeated.

-It is unclear what the alpha’s are in lines 325-326. If these are Cronbach’s alphas related to the consistency of the scale in this study, suggest this be presented in Results (with appropriate details in the Methods).

-First, this sentence: ‘A receiver operator curve (ROC)… in initiating MOUD’ (lines 333-334) should be below in the Stats Analysis section. But, more importantly, this approach is missing critical details such as what is the outcome used, as the ROC is classifying a dichotomous state/outcome (positive/negative for something), and what was the purpose of this ROC (what purpose does it play in the research question).

-Then, the next sentence ‘The AUC was 0.8554… Appendix).’ should be in the Results.

-Suggest the following changes for the Stats Analysis section: 1) Clearly state and define primary and secondary outcomes (e.g., ‘Our primary outcome was the continuous interest score reported by participants at [what visit(s)]. Secondary outcomes were initiation of MOUD defined as X.’), 2) the scientific question(s) alongside the analysis approach to answer this question (e.g., ‘To examine the intervention’s impact on interest in MOUD, we compared an individual’s pre and post interest score using X…’, 3) then, mention significance level and stats software. In summary, each analysis, including descriptive exploratory data analyses, need to link back to a scientific question, usually with an outcome/exposure(s) (or dependent/independent variables) (e.g., ‘To understand how those who initiated MOUD differed from those who did not, we compared baseline socio-demographic, X, and X characteristics. Chi-squared and Fishers…’).

-Suggest authors describe how they picked predictive variables of MOUD initiation and what they were.

RESULTS

-‘Forty-seven (39.5%) overdosed in the last 6 months.’ (lines 357-358) – it is unclear why this sentence is here as it seems better in ‘Participant characteristics.’

-‘Those who started MOUD were significantly more likely to be older, women, not have children, HIV, HCV or syphilis, relative to those who did not start MOUD.’ – this seems off relative to what is presented in Table 1. Those who initiated MOUD were more likely to HAVE children, be living with HIV and HCV, and LESS likely to be RPR positive. Suggest this sentence be re-written for clarity.

-‘Using the cutoff of >=8’ – 8 what exactly? It seems the authors dichotomized the MOUD interest score maybe and analyzed in this manner but there are no details of this in the Methods. This should be clearly detailed in the Methods. All analyses described in the Results section should be described in the Methods section

-‘The mean interest in MOUD score increased…’ – the analysis section does not mention the score being analyzed as a mean – rather the Wilcoxon signed-rank test was mentioned, which does not use the mean (it is a nonparametric test that uses ranking). This needs to be clarified in the Methods, Results, and Tables.

-For Tables 4 and 5 – the authors only refer to the tables without detailing or summarizing the results. Either these results should be presented in the text or excluded.

-Change ‘multivariate’ to ‘multivariable’ analysis. Additionally, mention all included covariates in the multivariable logistic regression (e.g., is this adjusted for age, MOUD interest score at baseline, treatment for other health conditions like HIV, depression, etc.)

-Why are details on OAT initiation bulleted? Suggest authors integrate this into the text.

TABLES/FIGURES

-Most table titles are too vague – for example, ‘participants’ should be replaced with who the participants are (e.g., people on probation with OUD in Moldova, 2019-2023).

-Table 1 – Dichotomous variables don’t need both yes and no rows, typically, as this makes the table long and difficult to read (for example, just a row for ‘partnered’ and ‘has children’).

-Table 1 – ‘Previously attempted to receive OAT’ – this is vague and unclear. Is this inclusive of people who tried but never started OAT as well as those with a history of using OAT? Suggest revision for clarity.

-Table 1 – Was treatment status collected from those with HIV, HCV, syphilis, depression? If so, suggest adding to the table.

-Table 1 – Details on some scales/instruments in this table are missing from the Methods and should be added – specifically DAST-10 and the ‘social support scale.’

-Figure 2 – ‘Interested in OAT-Pre/Post’ both seems to be presented twice, and I see the note in the legend but think this can be better presented without having to go into the legend – suggest a revision. The ‘brief intervention’ line also seems to be in the wrong place?

DISCUSSION

-Much of the detailed discussion on how SBIRT could be implemented in probation settings seems out of scope as the present study did not examine implementation outcomes. Similarly, the paragraphs on dynamics of prison and probation settings seem to talk about hypotheses but none of which are related to the findings of the present study. Suggest the authors start with what the objectives of the study were, then what the findings were, putting the findings in the Moldovan context and with other literature, and briefly, implications for next steps and public health/probation programs.

-‘brief interventions have demonstrated little benefit related to drug use’ – the evidence on motivational interviewing is certainly mixed but would recommend not concluding ‘little benefit’ for all brief interventions: https://pmc.ncbi.nlm.nih.gov/articles/PMC10714668/

-‘ In contrast to both these studies that demonstrated the utility of SBIRT for improving MOUD initiation…’ – this sentence is confusing. Which studies are the authors are referring to? As it cannot be the present and sister study in Ukraine since all participants received the SBIRT and thus its impact on MOUD initiation could not be assessed. Rather, the present and Ukraine study assess the impact of SBIRT on MOUD interest (NOT initiation).

-While the pre/post design of the study was an appropriate choice to understand the impact the motivational interviewing on MOUD interest, the authors don’t mention the clear limitations of this design versus an RCT with a control group. This should be added.

-It is a bit odd for the authors to note that they did not look at the influence of the national registry on MOUD initiation since that was a reason participants could select for not initiating (Table 6). Suggest a revision to clarify what exactly they see as the limitation.

-The findings do not support this sentence: ‘…this modified SBIRT strategy deployed in probation by trained interventionists was effective at improving the uptake of MOUD.’ Unless there is something in the study design that is not currently detailed, the study did not and could not examine the impact of the SBIRT on MOUD uptake as all participants received the SBIRT intervention – there is no comparison group. The pre/post measurement of MOUD interest assessed changes in MOUD interest associated with the motivational interviewing but not uptake.

7. PLOS authors have the option to publish the peer review history of their article (what does this mean?). If published, this will include your full peer review and any attached files.

**Do you want your identity to be public for this peer review?** For information about this choice, including consent withdrawal, please see our Privacy Policy.

Reviewer #1: No

Reviewer #2: **Yes:**Allison M. McFall

Figure Resubmissions:

---

## [Decision Letter · Decision Letter 2]

8 Dec 2025

PGPH-D-24-02730R2

Probation as a targeted entry point for scaling up opioid agonist therapies in Moldova using a modified screening, brief intervention and referral to treatment strategy

Dear Dr. Ponticiello,

Thank you for submitting your manuscript to PLOS Global Public Health. After careful consideration, we feel that it has merit but does not fully meet PLOS Global Public Health’s publication criteria as it currently stands. Therefore, we invite you to submit a revised version of the manuscript that addresses the points raised during the review process.

The manuscript has been evaluated by two reviewers, and their comments are available below.

The reviewers are generally positive but are requesting revisions to improve clarity, particularly relating to the statistical analysis.

Could you please carefully revise the manuscript to address all comments raised?

We look forward to receiving your revised manuscript.

Kind regards,

Alejandro Torrado Pacheco, PhD

Staff Editor

Journal Requirements:

1. Please provide additional information regarding the considerations made for the participants included in this study. For instance, please discuss whether participants were able to opt out of the study and whether individuals who did not participate receive the same treatment offered to participants.

2. Please include a complete copy of PLOS’ questionnaire on inclusivity in global research in your revised manuscript. Our policy for research in this area aims to improve transparency in the reporting of research performed outside of researchers’ own country or community. The policy applies to researchers who have travelled to a different country to conduct research, research with Indigenous populations or their lands, and research on cultural artefacts. The questionnaire can also be requested at the journal’s discretion for any other submissions, even if these conditions are not met. Please find more information on the policy and a link to download a blank copy of the questionnaire here: https://journals.plos.org/globalpublichealth/s/best-practices-in-research-reporting. Please upload a completed version of your questionnaire as Supporting Information when you resubmit your manuscript.

Additional Editor Comments (if provided):

Reviewers' comments:

Reviewer's Responses to Questions

**Comments to the Author**

1. If the authors have adequately addressed your comments raised in a previous round of review and you feel that this manuscript is now acceptable for publication, you may indicate that here to bypass the “Comments to the Author” section, enter your conflict of interest statement in the “Confidential to Editor” section, and submit your "Accept" recommendation.

Reviewer #1: (No Response)

Reviewer #2: (No Response)

2. Does this manuscript meet PLOS Global Public Health’s publication criteria? Is the manuscript technically sound, and do the data support the conclusions? The manuscript must describe methodologically and ethically rigorous research with conclusions that are appropriately drawn based on the data presented.

Reviewer #1: (No Response)

Reviewer #2: Partly

3. Has the statistical analysis been performed appropriately and rigorously?

Reviewer #1: (No Response)

Reviewer #2: No

4. Have the authors made all data underlying the findings in their manuscript fully available (please refer to the Data Availability Statement at the start of the manuscript PDF file)?

Reviewer #1: (No Response)

Reviewer #2: No

5. Is the manuscript presented in an intelligible fashion and written in standard English?

Reviewer #1: (No Response)

Reviewer #2: Yes

6. Review Comments to the Author

Reviewer #1: (No Response)

Reviewer #2: Thank you for the opportunity to review the revised manuscript entitled ‘Probation as a targeted entry point for scaling up opioid agonist therapies in Moldova using a modified screening, brief intervention and referral to treatment strategy.’ I appreciate all the time and effort the authors put in to address my prior comments – I recognize there were many. Below are remaining comments that should be addressed to clarify the aims, methods, and conclusions that can be drawn from the work.

Comment 6:

The authors mention that one of primary goals of the study was to pilot test the feasibility of targeting probation as an opportunity to encourage MOUD, but it isn’t clear after reading the paper how ‘feasibility’ is defined or assessed. Suggest the authors more clearly define what their aims are for this paper.

Response: We have left the sentence as is to reflect this was indeed a feasibility study although implementation outcomes, specifically, were not collected.

Reviewer response: I will continue to push on this point. As the paper is laid out now, noting that a primary goal of the research is assessing feasibility is misleading as there are no results currently that address feasibility. What does feasibility mean in the context of THIS work?

Comment 26:

First, this sentence: ‘A receiver operator curve (ROC)... in initiating MOUD’ (lines 333-334) should be below in the Stats Analysis section. But, more importantly, this approach is missing critical details such as what is the outcome used, as the ROC is classifying a dichotomous state/outcome (positive/negative for something), and what was the purpose of this ROC (what purpose does it play in the research question).

Response: We have moved the ROC analysis to the Statistical Analysis section and expanded the description to specify that the ROC was used to determine the optimal cutoff point on the 0–10 MOUD interest scale for predicting subsequent MOUD initiation. The area under the curve (AUC = 0.8554) identified a score of 8 as the most discriminative threshold and this was used in subsequent analyses. (Lines 388 – 390)

Reviewer response: The revised text in the manuscript remains unclear; the above explanation which includes what the ‘gold standard’ is (i.e., documented MOUD initiation among enrolled study participants) is better and should be presented as such in the manuscript. Since the interest score was collected twice from each participant, were all of these data points used in the AUROC or just the first? Suggest detailing this process a bit more so readers can follow what was done.

Comment 28:

Suggest the following changes for the Stats Analysis section:

1. Clearly state and define primary and secondary outcomes,

2. Link each analysis to a scientific question,

3. Mention significance level and stats software.

Response: We thank the reviewer for their suggestion but have left the stats section as is.

Reviewer response: I will continue to push on this point as well. The section is difficult to follow and should be revised for clarity.

Comment 33:

‘The mean interest in MOUD score increased...’ – the analysis section does not mention the score being analyzed as a mean – rather the Wilcoxon signed-rank test was mentioned. Clarify in Methods, Results, and Tables.

Response: We revised the text to indicate that mean values were presented descriptively for interpretability, while statistical significance for paired differences was determined using the Wilcoxon signed-rank test. This clarification has been applied consistently in the Methods, Results, and table legends. (Lines 396 – 398).

Reviewer response: This approach – presenting a summary statistic that is not used in the statistical hypothesis test – is confusing and unnecessary. Conclusions such as this in the Discussion is not accurate: “Mean interest scores in starting MOUD increased significantly after the brief intervention.” Presentation of the median would be appropriate. Suggest revision.

Comment 42:

Figure 2 – ‘Interested in OAT-Pre/Post’ appears twice and the ‘brief intervention’ line seems misplaced. Suggest revision.

Response: We thank the reviewer for their notes. The previous reviewer provided no additional comments on the figures so we will maintain them as is unless the editors request otherwise.

Reviewer response: This is an excellent example of why more than 1 (or 2) reviewers is so helpful to the process! It would behoove the authors to take some time to review the figure for accuracy (i.e., why would the brief intervention come AFTER the ‘post OAT interest assessment’?) and fix before publication.

Comment 48:

The findings do not support this sentence: ‘...this modified SBIRT strategy deployed in probation by trained interventionists was effective at improving the uptake of MOUD.’ Revise to reflect actual findings (change in interest, not uptake).

Response: We revised the sentence to more accurately reflect our results:

“This modified SBIRT strategy deployed in probation by trained interventionists was effective in identifying individuals with OUD and increasing their interest in MOUD. The intervention also resulted in a substantial increase in the number of people with OUD initiating MOUD relative to before the intervention” (Line 1088 - 1091)

Reviewer response: It is not clear from the Results, how the authors reach the conclusion that ‘The intervention also resulted in a substantial increase in the number of people with OUD initiating MOUD relative to before the intervention.’ What estimate are the authors comparing to? By design, none were on MOUD at baseline (i.e., an inclusion criterion was not currently on MOUD), – so are authors comparing to 0? Suggest revising Results and/or Conclusions to be clear on this point.

7. PLOS authors have the option to publish the peer review history of their article (what does this mean?). If published, this will include your full peer review and any attached files.

**Do you want your identity to be public for this peer review?** For information about this choice, including consent withdrawal, please see our Privacy Policy.

Reviewer #1: No

Reviewer #2: **Yes:**Allison M. McFall

Figure Resubmissions:

---

## [Decision Letter · Decision Letter 3]

9 Feb 2026

PGPH-D-24-02730R3

Probation as a targeted entry point for scaling up opioid agonist therapies in Moldova using a modified screening, brief intervention and referral to treatment strategy

Dear Dr. Ponticiello,

Thank you for submitting your manuscript to PLOS Global Public Health. After careful consideration, we feel that it has merit but does not fully meet PLOS Global Public Health’s publication criteria as it currently stands. Therefore, we invite you to submit a revised version of the manuscript that addresses the points raised during the review process.

The manuscript has been further evaluated and the reviewer comments are available below.

Could you please carefully revise the manuscript to address all comments raised?

We look forward to receiving your revised manuscript.

Kind regards,

Ilse Bloom

Staff Editor

Journal Requirements:

Additional Editor Comments (if provided):

Reviewers' comments:

Reviewer's Responses to Questions

**Comments to the Author**

1. If the authors have adequately addressed your comments raised in a previous round of review and you feel that this manuscript is now acceptable for publication, you may indicate that here to bypass the “Comments to the Author” section, enter your conflict of interest statement in the “Confidential to Editor” section, and submit your "Accept" recommendation.

Reviewer #1: (No Response)

Reviewer #2: (No Response)

2. Does this manuscript meet PLOS Global Public Health’s publication criteria? Is the manuscript technically sound, and do the data support the conclusions? The manuscript must describe methodologically and ethically rigorous research with conclusions that are appropriately drawn based on the data presented.

Reviewer #1: (No Response)

Reviewer #2: Partly

3. Has the statistical analysis been performed appropriately and rigorously?

Reviewer #1: (No Response)

Reviewer #2: No

4. Have the authors made all data underlying the findings in their manuscript fully available (please refer to the Data Availability Statement at the start of the manuscript PDF file)?

Reviewer #1: (No Response)

Reviewer #2: Yes

5. Is the manuscript presented in an intelligible fashion and written in standard English?

Reviewer #1: (No Response)

Reviewer #2: Yes

6. Review Comments to the Author

Reviewer #1: (No Response)

Reviewer #2: Thank you for the opportunity to review the revised manuscript entitled ‘Probation as a targeted entry point for scaling up opioid agonist therapies in Moldova using a modified screening, brief intervention and referral to treatment strategy.’ I appreciate all the time and effort the authors put in to address my prior round of comments.

One last comment on the comparison of MOUD scores – pre and then post-intervention – using the Wilcoxon signed-rank test for paired data. In the Abstract and main paper, the authors note that ‘MOUD interest did not increase significantly after the brief intervention although post-intervention scores were shifted toward higher values overall (Wilcoxon signed-rank test, p<0.001).’ It looks that the median/IQR at both time points is 5 (3-7) but the test p-value was <0.001 meaning there is a significant difference in the median score? Is that correct? The data presented and p-value seem to conflict so suggest the authors clarify and, if needed, update inferences/conclusions, made based off of this comparison.

7. PLOS authors have the option to publish the peer review history of their article (what does this mean?). If published, this will include your full peer review and any attached files.

**Do you want your identity to be public for this peer review?** For information about this choice, including consent withdrawal, please see our Privacy Policy.

Reviewer #1: No

Reviewer #2: **Yes:**Allison M. McFall

Figure Resubmissions:

---

## [Decision Letter · Decision Letter 4]

1 Apr 2026

Probation as a targeted entry point for scaling up opioid agonist therapies in Moldova using a modified screening, brief intervention and referral to treatment strategy

PGPH-D-24-02730R4

Dear Mr. Ponticiello,

We are pleased to inform you that your manuscript 'Probation as a targeted entry point for scaling up opioid agonist therapies in Moldova using a modified screening, brief intervention and referral to treatment strategy' has been provisionally accepted for publication in PLOS Global Public Health.

Best regards,

Julia Robinson

Executive Editor

Reviewer Comments (if any, and for reference):

Reviewer's Responses to Questions

**Comments to the Author**

1. If the authors have adequately addressed your comments raised in a previous round of review and you feel that this manuscript is now acceptable for publication, you may indicate that here to bypass the “Comments to the Author” section, enter your conflict of interest statement in the “Confidential to Editor” section, and submit your "Accept" recommendation.

Reviewer #1: (No Response)

Reviewer #2: All comments have been addressed

2. Does this manuscript meet PLOS Global Public Health’s publication criteria? Is the manuscript technically sound, and do the data support the conclusions? The manuscript must describe methodologically and ethically rigorous research with conclusions that are appropriately drawn based on the data presented.

Reviewer #1: (No Response)

Reviewer #2: Yes

3. Has the statistical analysis been performed appropriately and rigorously?

Reviewer #1: (No Response)

Reviewer #2: Yes

4. Have the authors made all data underlying the findings in their manuscript fully available (please refer to the Data Availability Statement at the start of the manuscript PDF file)?

Reviewer #1: (No Response)

Reviewer #2: Yes

5. Is the manuscript presented in an intelligible fashion and written in standard English?

Reviewer #1: (No Response)

Reviewer #2: Yes

6. Review Comments to the Author

Reviewer #1: (No Response)

Reviewer #2: (No Response)

7. PLOS authors have the option to publish the peer review history of their article (what does this mean?). If published, this will include your full peer review and any attached files.

**Do you want your identity to be public for this peer review?** For information about this choice, including consent withdrawal, please see our Privacy Policy.

Reviewer #1: No

Reviewer #2: **Yes:**Allison M. McFall
